# A Global Perspective on Microplastic Occurrence in Sediments and Water with a Special Focus on Sources, Analytical Techniques, Health Risks, and Remediation Technologies

Venkatraman Nagarani Prapanchan [1], Erraiyan Kumar [2], Thirumalaisamy Subramani [1,*], Udayakumar Sathya [3] and Peiyue Li [4,5]

[1] Department of Geology, College of Engineering Guindy (CEG), Anna University, Chennai 600025, India; prapanchanau3131@gmail.com
[2] Department of Mining Engineering, College of Engineering Guindy (CEG), Anna University, Chennai 600025, India; kumarmine05@gmail.com
[3] Department of Environmental Engineering, CSIR-Central Leather Research Institute, Chennai 600020, India; sathcivil@gmail.com
[4] School of Water and Environment, Chang'an University, Xi'an 710064, China; lipy2@163.com
[5] Key Laboratory of Subsurface Hydrology and Ecological Effects in Arid Region of the Ministry of Education, Chang'an University, Xi'an 710064, China
* Correspondence: geosubramani@gmail.com; Tel.: +91-44-2235-7770

**Abstract:** Microplastics have become so pervasive that they seem to be present all around us due to the significant environmental threat they pose. Microplastic pollutants have become an issue as global plastic manufacturing has increased. Microplastics are plastic wastes with particulates less than 5 mm in size that are absorbed by sediment, water, the atmosphere, and living beings before affecting health. Moreover, there is a shortage of knowledge on the distribution, sources, toxic effects, analytical techniques, and removal technologies of microplastics. This review examines the distribution and global abundance of microplastics in aquatic and terrestrial environments, analytical methods, remediation technologies, and health risks. The following are included in this review article: (1) sampling, extraction, and analysis techniques for microplastics in sediment, water, and salt; (2) the source, global distribution, and concentration of microplastics; (3) toxicity and consequences of microplastics on human health; and (4) several methods for removing microplastics, grouped into three categories: engineered, biopolymer, and bioengineered approaches. The worldwide distribution, identification, toxic effects, and remediation technology of microplastics will benefit greatly from this review.

**Keywords:** microplastic; global distribution; toxicity; source; remediation technology

## 1. Introduction

Plastics have become an integral part of modern society because of their versatility, durability, and affordability. However, their widespread use has led to the proliferation of microplastics, defined as plastic particles smaller than 5 mm. Figure 1 shows the size-based classification of plastic particles proposed by different authors. Microplastics (MPs) are ubiquitous in the ecosystem and have been found in land, oceans, freshwater systems, and air. Microplastics are tiny plastic particles that are less than 5 mm in size and are becoming a significant environmental concern owing to their persistence, abundance, and potential impact on ecosystems and human health [1,2]. They are produced both intentionally, such as in personal care and cleaning products, and unintentionally through the fragmentation of larger plastic items that can come from various sources, including cosmetic products, clothing, and packaging materials [1]. There is growing environmental concern due to the widespread distribution of microplastics in the environment and their potential ecological and human health impacts. The persistence of microplastics in the environment, combined with their ability to absorb and transfer toxic

contaminants, raises significant concerns about their impact on wildlife, ecosystems, and human health [2]. They are widely distributed in the environment, including land, oceans, freshwater systems, and air. In recent years, increasing research has been conducted on the occurrence and distribution of microplastics in various environmental media, including seawater, sediments, soil, and biotic tissues [3,4]. These studies have revealed the presence of microplastics in virtually every corner of the planet, including remote areas such as the Arctic and deep sea [5]. Despite the growing body of research on microplastics, many uncertainties and knowledge gaps remain, particularly with regard to the ecological and human health impacts of these particles [6]. Furthermore, there is a lack of effective measures to reduce the release of microplastics into the environment, as well as limited capacities for removing them once they have entered the environment [1]. Several studies have been conducted to determine the sources of microplastics in the environment. The main sources of microplastics are the breakdown of large plastic debris and the discharge of microplastics from various industries, including the cosmetic and textile industries. Other sources include the release of microfibers from synthetic textiles during washing and the release of microbeads from personal care products, such as toothpaste and exfoliating scrubs [3]. One of the key challenges in understanding the sources of microplastics is the complex and diverse pathways through which they can enter the environment. According to a study by ENSSER [7], microplastics can enter the environment through various pathways, including wastewater discharge, atmospheric deposition, and the transfer of plastic debris from land to sea through stormwater runoff. The occurrence and distribution of microplastics have been studied globally in recent years, and research suggests that they are ubiquitous and pervasive. In a study by Koelmans et al. (2017), microplastics were found in all major oceanic gyres and in all samples collected from the Arctic to the Antarctic [8]. Similarly, another study by Eerkes et al. (2019) found that microplastics in lakes, rivers, and groundwater were present in all freshwater systems studied [9]. The impact of microplastics on the environment and biota is not yet fully understood, but laboratory and field studies have shown that they can have toxic effects on a variety of species, including marine mammals, birds, and fish [10]. Moreover, microplastics have been shown to adsorb toxic pollutants from the surrounding environment, making them even more hazardous to wildlife and human health [11].

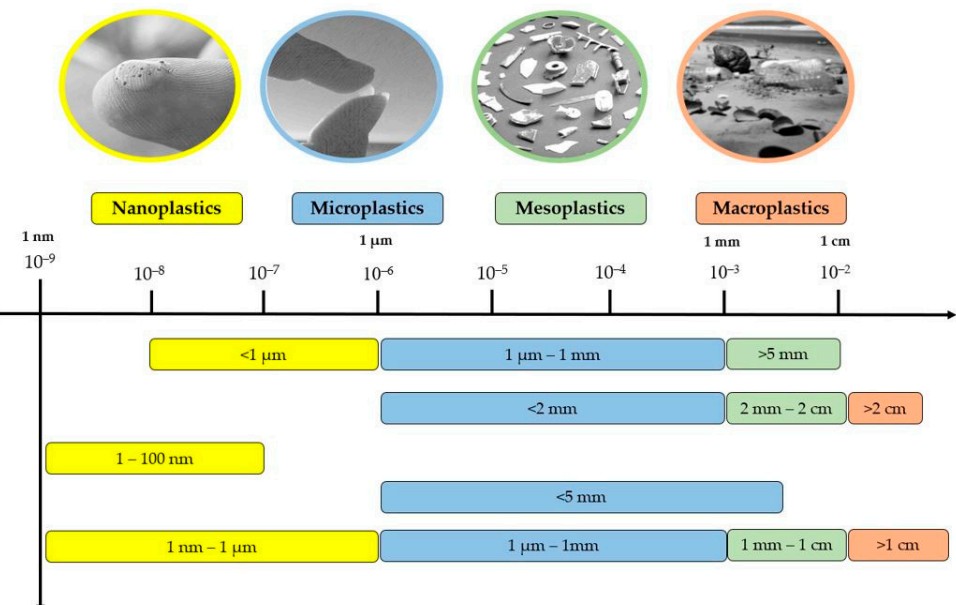

**Figure 1.** Different authors' stated definitions of plastics based on size [1,2].

In marine ecosystems, microplastics can negatively affect species at all trophic levels, from phytoplankton to top predators. For example, Anbumani et al. (2018) found that microplastics are ingested by a wide range of marine organisms, including zooplankton, fish, and seabirds, which can lead to physical harm and impaired feeding and digestion [12]. Another study by Prata et al. (2020) found that microplastics can also negatively impact the growth and reproduction of intertidal invertebrates [13]. The impact of microplastics on terrestrial ecosystems is less well understood, but recent studies have shown that they can have negative effects on soil biota and plant growth [12]. For example, Prata et al. (2019) found that exposure to microplastics reduced the growth and reproduction of earthworms, which can have a cascading effect on soil structure and nutrient cycling [13].

Microplastics pose a significant threat to marine wildlife and their ecosystems, as they can be ingested by organisms and absorb toxic pollutants, leading to adverse impacts on their health and the food chain. Mitigation of microplastics requires a multifaceted approach, including reducing the production and release of microplastics through regulation, as well as increasing public awareness and education. In 2011, the United States became the first country to ban the use of microbeads in personal care products, followed by several other countries [14]. The implementation of extended producer responsibility policies, which hold manufacturers responsible for end-of-life management of their products, can also contribute to reducing microplastic pollution [14]. In addition to reducing the production of microplastics, effective waste management practices, including proper disposal and recycling, are crucial for mitigating their release into the environment. The implementation of waste management systems, such as mechanical and biological treatments, can reduce the amount of microplastics entering the environment [15]. The use of biodegradable alternatives, such as bioplastics, can also contribute to reducing the amount of microplastics in the environment [16]. The effective management of microplastics also requires monitoring and assessment efforts to better understand the distribution, fate, and impact of microplastics in the environment. This includes monitoring water bodies, sediment, and biota, as well as developing and improving analytical methods for the detection and quantification of microplastics [17,18]. Despite ongoing efforts, microplastics remain a significant challenge, and continued research and action are necessary to address this issue. Further research is needed to assess the full extent of the impact of microplastics on the environment and human health, as well as the efficacy of various mitigation and management strategies.

The potential health risks associated with exposure to microplastics have become a growing concern among the scientific community and a pressing public health issue. Microplastic exposure can occur through various pathways, including inhalation, ingestion, and dermal contact. Once in the body, microplastics can interact with biological systems and potentially cause harm. Several studies have shown that microplastics contain toxic chemicals, such as bisphenol A (BPA) and phthalates, which can leach into the environment and pose a threat to human health [11]. Moreover, microplastics can act as carriers of persistent organic pollutants (POPs), which have been linked to numerous health problems, including cancer, endocrine disruption, and reproductive and developmental toxicity [14]. The ingestion of microplastics has become a particularly concerning issue, as they have been found in a variety of food sources, including fish, shellfish, and table salt. Once ingested, microplastics can remain in the digestive system for extended periods, potentially leading to the release of toxic chemicals into the body. Furthermore, microplastics have been shown to impair digestive and absorption processes in some species, leading to decreased nutrient uptake and energy utilization [14]. In addition to oral exposure, inhalation of microplastics is a growing concern. Microplastics are found in both indoor and outdoor air, with higher concentrations observed in urban areas. When inhaled, microplastics can reach the respiratory system and potentially cause harm. Some studies have suggested that microplastic inhalation may contribute to the development of respiratory and cardiovascular problems, although further research is needed to fully understand their health

effects [19]. Srihari et al. (2022) provided sources of density of plastic products in aquatic and terrestrial environments as shown in Table 1 [19].

**Table 1.** Density of plastic products in aquatic and terrestrial environments [16–19].

| Polymer Types | Products | Density of Plastic Material (g/cm$^{-3}$) |
| --- | --- | --- |
| Polymethyl methacrylate (PMMA) | shatterproof windows | 1.09–1.20 |
| Polyvinylchloride (PVC) | pipes, electric cables, clothing | 1.16–1.58 |
| Polyester (PES) | polyester clothes | 1.24–2.3 |
| Polytetrafluoroethylene (PTFE) | slide plates, seals, gaskets | 2.1–2.3 |
| Low-density polyethylene (LDPE) | plastic bags, squeeze bottles | 0.89–0.93 |
| High-density polyethylene (HDPE) | detergent bottles | 0.94–0.98 |
| Polypropylene (PP) | clothing, stoppers | 0.83–0.92 |
| Polyethylene terephthalate (PET) | water bottles | 0.96–1.45 |
| Polyamide (PA) | textile, tooth brush | 1.02–1.16 |
| Polystyrene (PS) | ready to-eat food | 1.04–1.1 |

In light of these concerns, this study aimed to contribute to the growing body of knowledge on microplastics by conducting a comprehensive analysis of their source, occurrence, distribution, sampling, identification techniques, recommendations, remediation technologies, and possible health risks associated with microplastic ingestion [19]. The impact of microplastics on marine and terrestrial ecosystems is a growing concern that requires continued research to better understand the extent and nature of their effects. The results of this study provide valuable information for the development of effective management strategies to reduce the release of microplastics into the environment and minimize their impact. Mitigation and management of microplastics require a collaborative and multidisciplinary approach, including reducing the production of microplastics, improving waste management practices, and increasing monitoring and assessment efforts. Overall, the available evidence suggests that exposure to microplastics poses a significant risk to human health. However, much of the research in this area is still in its early stages, and more studies are needed to fully understand the extent of health risks associated with microplastics. Nevertheless, it is clear that the proliferation of microplastics in the environment is a pressing public health issue, and immediate action must be taken to reduce exposure and minimize risks.

## 2. Data Collection

To conduct the literature review, we searched through various databases, including ScienceDirect, Scopus, PubMed, and ISI Web of Science. The key terms were employed to find scientific articles (published up until 30 December 2022) were: "microplastic(s)", "plastic debris", "global", "India", "Ocean", "sediment", "water", "salt", "terrestrial", and "freshwater". After screening the publications by research region, publications that focused on global and Indian environmental matrices, as well as beaches, estuaries, offshore areas, lakes, rivers, and oceans, were chosen. For this study, we examined a collection of 100 research articles. Our aim was to provide a condensed overview of the following topics: sample methodology, source, microplastic particle extraction procedures, detection techniques, microplastic concentration, characterization, recommendation, remediation technologies, and health risks. To study microplastic research, a review of several marine and terrestrial environmental systems, including islands, lagoons, beaches, seas, and estuaries, was conducted.

## 3. Types of MP Particles

### 3.1. Primary MPs

Primary microplastic particles are characterized as plastic products that are small. These plastics are commonly found in cosmetic products and face washes [20], and there is growing evidence that they are employed in pharmacies as drug delivery systems [21]. In the personal care and beauty goods industry, as well as the detergent industry, polyurethane main microplastics produced at least half of the micro-litter in 2017. Bath gels, shaving lubricants, eye makeup, personal care products, blushing powders, cosmetics foundation, eyeliner, shaving foam, infant goods, bubble bath creams, hair dye, nail polish, insect repellent, and moisturizers contain microplastics [22], according to Auta et al. (2017). The presence of pellets in this class has been criticized, and virgin plastic manufactured pellets are usually 2–5 mm in size. They can also be categorized as primary microplastics under the more expansive descriptions of microplastic particles [23]. Microplastic "scrubbers" have taken the role of previously popular organic ingredients such as powdered almonds, oats, and pumice in exfoliating hand washes and facial cleansers [24]. The use of exfoliating cleansers with plastic ingredients has increased significantly since microplastic scrubbers used in cosmetics were patented in the 1980s [25]. Gregory et al. discovered the presence of polystyrene spheres (<2 mm) and polyethylene and polypropylene particles (<5 mm) in one skincare product. Fendall and Sewell observed microplastics with erratic shapes in other beauty products. The microplastics had an average diameter of <0.5 mm [26]. The annual output of primary microplastic particles in the paint industry is 220 tons, based on a European Commission report [27].

### 3.2. Secondary MPs

Secondary microplastics are microscopic plastic particles that are produced as larger plastic garbage breaks down, both on land and in the sea [28,29]. The underlying stability of plastic trash can be compromised over time by a confluence of physical, chemical, and biological processes, leading to disintegration [30,31]. Long-term exposure to sunlight can cause photodegradation of plastics: the polymer structure oxidizes when exposed to the sun's UV radiation and induces depolymerization in polymers [32,33]. A consequence of this deterioration could be the leaching of plastics from compounds intended to increase durability and corrosion resistance [34]. Plastic waste on land is exposed directly to oxygen and sunshine, so it degrades swiftly, becoming brittle, cracking, and "yellowing" over time [35,36]. Most plastics are increasingly prone to disintegration as their structural stability deteriorates [36]. This cycle continues, with plastic particles eventually shrinking and reaching the size of microplastics [28]. It is believed that microplastics can further disintegrate into nanoplastics.

## 4. Source of MPs

Plastic pollution enters the environment because of poor human activity and/or uncontrolled waste disposal [36]. Microplastic particles are mostly sourced from the disintegration of macroplastic items and tiny plastic granules used as abrasive scrubbers in cleansing and personal hygiene goods. The primary source is industrial and household products, such as bathrooms, hand, body, and facial cleaning agents, beauty products, small beads used as scrubbers in laundry items, powder, and resin pellets, among others, as the basic thermostatic sector feedstocks and abrasive plastic beads utilized for ship cleaning [37–42]. The pellets or microbeads present in cosmetics and laundry detergents are examples of primary microplastics, which are developed only for consumer use. Secondary microplastics are produced when larger plastics break down into finer particles and are ultimately introduced into the environment [43,44]. Both forms of microplastic (primary and secondary) are found in high amounts in terrestrial and aquatic habitats. Every year, an astonishing amount of 245 tons of microplastic particles are created, which wind up in bodies of water where they are ingested by aquatic organisms and absorbed into their bodies and tissues [45,46]. The bulk of plastic waste enters the environment through

land-based production and processing facilities, sewage systems, and biosolid waste. Recreational activities also significantly contribute to the aquatic ecosystem plastic load, and their invasion across inland water bodies has not been properly examined. This could be the result of improperly managed landfill disposal, debris blown by the wind, storm surges, river transport, etc. Hence, the accumulation of plastics on land eventually enters oceanic and riverine ecosystems. The usage of microbeads in cosmetics and synthetic textile fibers may be an additional route for microplastic particles to reach the freshwater environment via the sewer system. Therefore, effective microplastic elimination through a wastewater treatment facility (STP) is crucial for decreasing pollutants in the environment. Maximum plastic rubbish patches are expected to be from densely inhabited areas in which a larger percentage of plastic items, such as containers, grocery bags, and hygiene supplies, are utilized [1]. The discharge of microplastic particle-sized fibers as a consequence of fabric washing has been extensively recorded [47–49]. This eventually makes its way into aquatic habitats, where it affects the organisms. The average consumption of polyethylene microplastic particles in liquid soap is calculated to be roughly 2.4 mg per person per day for the US population [50].

Tourism and fishing operations, industrial sewage discharge, urban runoff [51,52], and marine transportation operations are manmade sources of microplastic pollution [53]. The main element contributing to the introduction of microplastic particles into the aquatic environments in Poland and Germany is the passage of the Vistula and Oder streams into the Gulf of Gdansk and Pomeranian Bay. It is believed that riverine mobility is a significant route for microplastics to reach coastal habitats [54]. In Singapore, human activity such as fishery and recreational use of fishing lines, food containers, plastic water bottles, and plastic detergents containers are the major sources of microplastics. Plastic waste degrades as a result of wave action, weather exposure, and ultraviolet light; finally, microplastic particles end up in Singapore's mangroves [55]. Microplastic contaminants have been abundantly observed in places as distant as Antarctica, and their causes include proximity to the area's treatment facility for wastewater, ship traffic, coastal scientific research operations, and transfer by ocean circulation [56]. According to Stolte et al. (2015), city wastes, industrial output sites, the fishing industry, and tourist industry are the most likely sources of the higher microplastic concentration [57]. In addition to wastewater and effluent intake, Horton et al. (2017) identified highway markings composed of thermoplastic composite paint as contributors to microplastic particles in the River Thames Basin (UK) [58]. In accordance with various studies, sewage discharge has been identified as a key contributor to microplastic particles [59]. Owing to the inadequate eradication of microplastics during sewage treatment [60], WWTPs are recognized contributors of microplastics in freshwater bodies. The development of microplastics has also been linked to tourism. According to a study by Retama et al. (2016), December, when tourism is at its peak along the Southern Pacific coast of Mexico, saw a higher proportion of microplastics than April, when human impact was at its lowest [61].

Microplastics are continuously released into the atmosphere through various sources [62]. According to Gondalia et al. (2020), tire wear caused by driving accounts for the release of 3.5 to 10 metric tons of microplastics into the atmosphere each year in Paris [63]. Synthetic textiles, such as polyester and nylon, continuously release microplastic fibers during washing and wearing, contributing up to 71% of total microplastic particles in the atmosphere, as found by Shi et al. (2020) in Beijing [64]. Plastic production and disposal release 0.15 to 0.84 metric tons of microplastics into the atmosphere each year, according to Cincinelli et al. (2020) in Italy [65,66]. Agricultural activities, such as tilling and fertilizing, also release microplastics into the atmosphere, estimated to be 0.1 to 0.8 metric tons each year, according to Barducci et al. (2020) in Switzerland [67]. The sources and types of microplastic pollutants in terrestrial, aquatic, and atmospheric environments are shown in Figure 2 and Table 2.

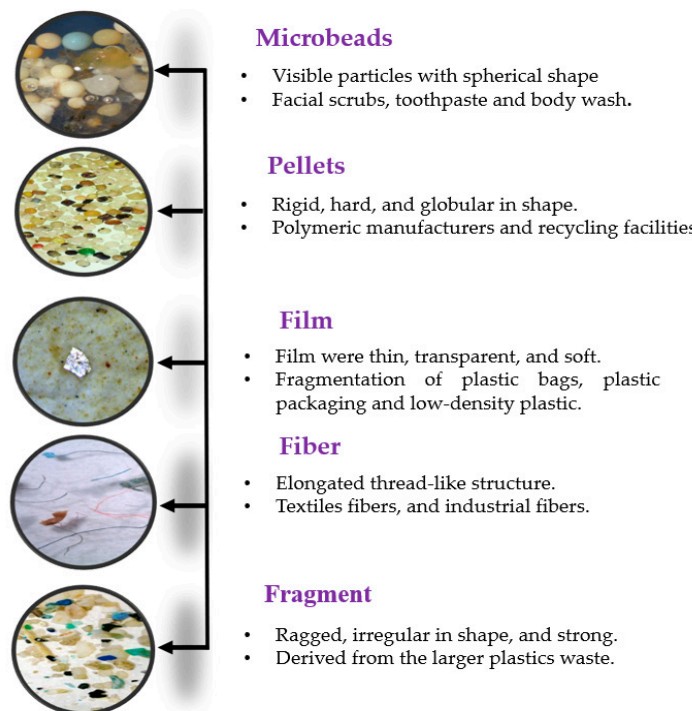

**Figure 2.** Sources and types of microplastic particles.

**Table 2.** Source of MP pollutants in the terrestrial, aquatic, and atmospheric environments [37–46].

| Source Types | Source of MPs Pollutants |
|---|---|
| Terrestrial sources | Improper waste management, composting methods in agriculture, indoor dust emissions from public buildings, use of treated wastewater from wastewater treatment plants, soil amendment in agriculture, digging of soil contaminated by plastics, construction sites, and building restoration, stormwater drain discharge, recreational boating, and freshwater recreation, use of plastic beads in medicine, municipal wastewater that has not been properly or adequately treated, industrial products, and process pollutants. |
| Water sources | Aquaculture and fishing, maritime, harbor, recreational boating, and beachside activities, maintenance of plastic-treated and plastic-painted maritime surfaces, offshore excavation and mining. |
| Atmosphere sources | Tire wear, synthetic textiles, plastic production and disposal, road dust, and agricultural activities. |

## 5. Identification and Extraction of Microplastics (MPs)

The sampling is characteristic of terrestrial and aquatic habitats, sample methodologies are crucial in determining the abundance and polymeric composition of microplastic particles [68–73]. The water, sediment, and salt samples are typically collected using basic environmental sampling methods [74–76].

### 5.1. Sampling

The sampling techniques utilized to obtain microplastic particles in sediment, water, and salt for global and Indian environments are given below.

#### 5.1.1. Sediment

Many studies have examined the prevalence of MPs in sediment from coasts, islands, lakes, and beaches. For manual grab methods for sediment, use implements such as hand shovels and metal spoons. Putting a steel or wooden structure on the surface of the sediment layer, pushing it downward to a depth of 1–5 cm, sweeping out the sediment material, and extracting the samples with a steel shovel, microplastics in sediments from lakes, coastal, and beach regions has typically been used [77]. Sampling of terrestrial and aquatic sediment

was performed using a Peterson grab sampler, and the various frame dimensions utilized in India and the global scenario were $25 \times 25$, $30 \times 30$, $50 \times 50$, $100 \times 100$, and $200 \times 200$ cm. The sampling depths ranged from 0 to 6 cm [78,79]. No research has been conducted to gather data on the vertical spatial variation of MP particles [80–82]. Microplastic-containing sediments have occasionally been collected manually or with steel tweezers [83–86]. The sample unit of microplastic analysis is generally items/kg [87–90] or items/g [91,92], with a few studies indicating items/m$^2$ [93–97] as their sampling unit.

5.1.2. Water

The prevalence and variation of microplastic particles throughout global coastal and freshwater environments were examined in several studies. Several different trawling and net designs were used, including bongo, manta, neuston, and plankton nets. A trawl net is often launched from a watercraft, immersed, and lifted in the direction of travel at a slow speed for a specified time or distance [97,98]. Multiplying the length of the tow by the width of the trawl yields the total area sampled. To gather samples from depths ranging between 20 cm and 2–5 m, manta trawling nets with mesh sizes of 112, 200, 300, and 335 μm were utilized. Eriksen et al. (2018) employed AVANI trawl instrument for microplastic particle sampling with rectangular entrance 60 cm $\times$ 14 cm and manta trawling net with a rectangular aperture 16 cm $\times$ 61 cm [98]. To determine the amount of water tested, a flow meter was fastened to trawl nets. Nonetheless, some studies did not indicate whether a flow meter was attached to the net, and if the net was not fully immersed or obstructed by an excessive amount of suspended and floating materials, simply measuring the distance during sampling could result in significant inaccuracies [99]. Trawl nets are commonly used to collect marine organisms for ecological studies, but they can also capture microplastics. Therefore, trawl nets have been increasingly used as a tool for sampling microplastics in the marine environment. A study by Lusher et al. (2017) investigated the efficiency of trawl nets in collecting microplastics from the water column. The study involved deploying a small-mesh trawl net off the coast of the UK and examining the microplastics collected. The results showed that trawl nets were effective in collecting microplastics, with an average of 1.9 microplastic particles per m$^3$ of water sampled. The study also found that microplastics were present in a range of sizes, with particles ranging from 20 μm to 5 mm in diameter [100]. Another study by Nelms et al. (2019) compared the efficiency of different sampling techniques for collecting microplastics in the marine environment, including trawl nets, plankton nets, and surface water pumps. The study found that trawl nets were the most efficient method for collecting microplastics, with the highest number of particles collected per unit of water sampled. However, using trawl nets for microplastic sampling can also present some limitations. One limitation is that trawl nets can potentially capture larger plastic debris, which may not accurately represent the abundance of microplastics in the water column. Additionally, trawl nets may not capture microplastics that are located deeper in the water column or that are associated with sediment [101]. To address these limitations, other studies have suggested using a combination of sampling techniques, including trawl nets, and plankton nets, to provide a more comprehensive picture of microplastic abundance in the marine environment [102]. Trawl nets are an effective tool for sampling microplastics in the marine environment, but they may not be the most suitable method in all situations. By combining different sampling techniques, researchers can obtain a more accurate picture of microplastic abundance in the water column. Trawl nets are commonly used in aquatic environments. However, they can potentially contaminate the samples with plastic particles. To minimize this risk, several strategies can be employed during sampling, including using alternative sampling techniques, cleaning and sterilizing nets, and avoiding areas with high levels of plastic pollution. One alternative sampling technique is the use of plankton nets, which are designed to collect smaller organisms and are less likely to accumulate plastic particles. Another option is to use gear with a smaller mesh size, which can reduce the amount of plastic that enters the net. Cleaning and sterilizing nets before use can also help to reduce

the potential for contamination. This can involve using a bleach solution or high-pressure hot water to remove any particles that may be present on the net. Avoiding areas with high levels of plastic pollution can also be an effective strategy. This may involve selecting sampling sites in areas where plastic pollution is less prevalent or avoiding sampling during periods when plastic is more likely to be present in the water column, such as after storms or during periods of high wind [99–101]. The process of grab sampling includes collecting water in a receptacle and filtering it on location. Additionally, a container with a set volume is submerged and filled with surface water for further analysis in a laboratory setting [102]. The microplastic concentrations in the water samples were measured in units of items/L, items/$km^2$, and items/$m^3$.

### 5.1.3. Salt

Five studies measured the prevalence of microplastic particles in commercial salt brands and natural salt pans that are accessible to customers. Samples of unprocessed sea salt for human use were collected in Bangladesh from specific natural salt pans and glass bottles with a clean label, and a capacity of 1 L was used to collect 500 g of salts from each location and brought to the laboratory [103]. Comparable analyses were conducted in Tuticorin on 14 salt brands made from borewells and saltwater, as well as 25 salt samples taken from various salt pans [83]. Seth and Shriwastav [86] employed eight commercial branded salts made in India. Kim et al. (2018) bought three commercialized kinds of sea salt in Indian stores [85].

### 5.2. MP Isolation Technique

In the majority of scientific literature, MPs were separated from sediment, water, and salt samples after sampling. Smaller microplastics were retrieved employing density separation and filtration techniques, whereas larger microplastics were assessed visually and extracted utilizing tweezers. Figure 3 shows the methods used for microplastic sampling and extraction in various environmental matrices. In laboratory settings, plastic items such as containers, pipettes, and tubes can release microplastics into the samples being tested, potentially contaminating the results. To avoid internal microplastic contamination, it is essential to follow proper handling and disposal procedures for plastic items. One of the key recommendations to avoid internal microplastic contamination is to use certified microplastic-free plastic items in laboratory settings. For example, the European Commission's Joint Research Centre has developed a certification scheme for microplastic-free plastic products used in laboratories. This certification scheme ensures that the plastic items have been tested and verified to be free of microplastics [104]. Another recommendation is to minimize the use of plastic items and opt for alternative materials when possible. Glass, stainless steel, and other non-plastic materials can be used as substitutes for plastic items. In addition, it is important to avoid unnecessary plastic packaging and to properly dispose of plastic items after use.

### 5.2.1. Sediment

Different-sized sieves were used to sieve dry sediment samples. With 63 mm [25], 300 mm [89,92,93], 0.1 mm [105], 1 mm [94,95,106], 2 mm [107], 3 mm [108], and 5 mm [31,109] are a few examples. Microplastics were removed from the sediment using the density separation technique, NaCl, and Zinc chloride. To break down the organic material, digestion with 30% $H_2O_2$ was performed either prior to or after density separation. With the digestion and density separation process, the sediment sample was left to dry naturally or in an oven before being passed through a range of mesh sizes of filter paper, including 0.4 mm [105], 0.45 mm [22,30], 0.8 mm [88,89,97], 1.2 mm [107], 38 mm [109] and 0.7 mm [20,87].

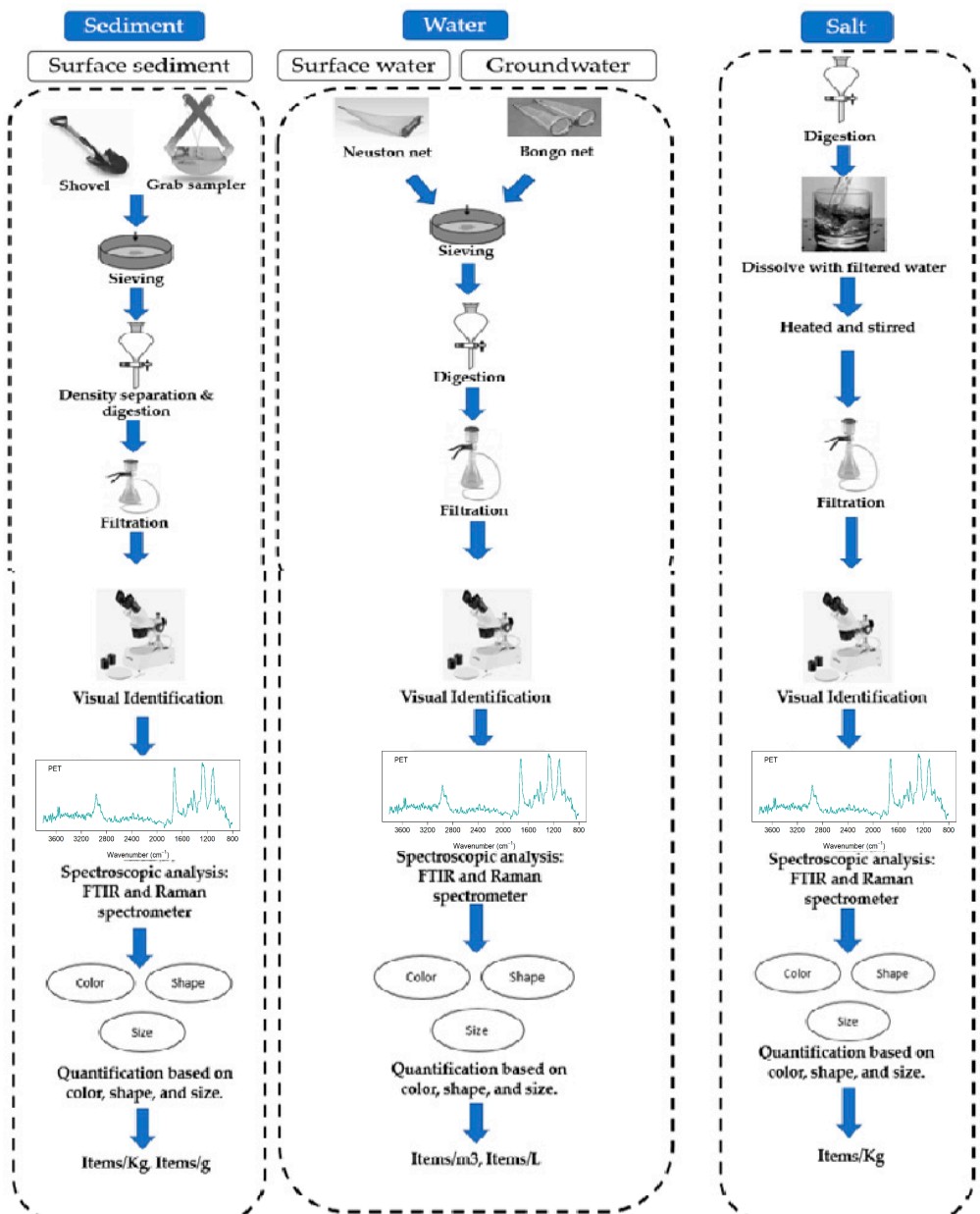

**Figure 3.** Methods for MP sampling and analysis in various environmental matrices are advised.

### 5.2.2. Water

In accordance with established techniques, collected water samples were filtered using a vacuum filtration setup and 0.45 mm Whatman cellulose filter paper before being dried at room temperature to determine the number of microplastics present [8,19] by Koelmans et al. (2019). In all investigations, the obtained water samples were either filtered or sieved to select the desired size. NaCl [87,93], ZnCl [19], and NaI [83,88] were employed for density separation to isolate microplastics from the water samples, and 30% $H_2O_2$ digestion was utilized to eliminate organic material.

### 5.2.3. Salt

To disintegrate any organic matter, 100 mL of 30% $H_2O_2$ was added to each sample in clean glass flasks before heating in a bath maintained at 65 °C for 24 h [75,110]. Each container received approximately 1000 mL of filtered water and the salt was stirred with a glass rod until it was completely dissolved. Using a vacuum setup, the resulting salt solution was quickly filtered with a cellulose membrane filter (0.2 mm, 0.45 mm, 0.8 mm,

and 2.7 mm pore size) [83,85,86]. The filter was positioned inside a petri dish made of glass and left to dry naturally at the surrounding temperature.

## 6. MP Detection, Chemical Categorization, and Quantification

### 6.1. Visual Examination

One of the first methods for quantifying microplastics in different environmental matrices is visual examination with the unaided eye, or with a stereoscope or microscope. Smaller microplastics require additional microscopic examination, whereas larger microplastics can be immediately separated. Depending on the homogeneity in color, brightness, and lack of cellular features, microplastics can be identified visually [86]. Several studies have attempted visual inspection along with hot-needle testing to validate the existence of plastics [83,89,97,111]. The visual examination method includes criteria for microplastics that have no visible organic structure: the colored particles had a consistent color throughout and were not mixed in with segmented fibers that had a flat ribbon-like appearance. Additionally, plastics were identified as particles that melted upon contact with a hot needle [83]. Microplastics are categorized visually by size, color, and shape, which enables one to determine their origin [87,88,111]. Although counting microplastics visually can save time, depending on the size range of the plastics, it is prone to error to count nonplastic particles as plastic and can also result in either extreme under-or overestimations of plastic materials.

### 6.2. Fourier-Transform Infrared Spectroscopy (FT-IR)

The most frequent technique for identifying and measuring microplastics is Fourier-transform infrared spectroscopy [112]. The investigation and characterization of microplastics have a long history with the FTIR spectroscopy approach, which provides the opportunity for precise polymer-type identification based on distinctive molecular fingerprint spectra. The FTIR method was employed in 80% of the investigations to determine the polymer types of microplastics in the various environmental matrices. The most frequently employed FTIR spectral range in the microplastics study was the mid-infrared band (400–4000 $cm^{-1}$). The most common spectroscopic modes are transmission and attenuated total reflection (ATR). To describe large MPs (>500 μm) in sediment and water, the ATR-FTIR approach was applied [86]. The technique of m-FTIR imaging or chemical imaging, which involves the use of FTIR and a confocal microscope, was utilized to determine the polymer varieties of small MPs (less than 500 μm) found in sediment, water, and salt [83,86]. The FTIR approach was used to analyze the weathering pattern in addition to the detection and classification of microplastics [78,112]. Despite its potential for identifying the types of polymers present in microplastics, FTIR has some drawbacks: the FTIR spectra for microplastics obtained through various modes do not match, making it crucial to research how chemical degradation affects the FTIR spectral bands of plastic materials prior to microplastic identification. To keep the particles during spectrum collection, a substrate is needed, but the spectroscopic interference caused by the addition of a substrate filter has not been fully addressed. Owing to refractive errors, irregularly shaped small microplastics would create unintelligible FTIR spectra. Moreover, FTIR is particularly effective in detecting the presence of water, generating wide peaks beyond 3000 $cm^{-1}$. Therefore, sample preparation was necessary before the measurements [113]. The FTIR method is a powerful tool for the identification of plastics. It works by measuring the absorption of infrared radiation by the chemical bonds in the plastic and comparing this to a reference spectrum in a database of known plastics. However, there are limits to the accuracy of this method, particularly when it comes to identifying altered plastics. One limitation of the FTIR method is that the plastic database provided by the manufacturer may not include information on the spectra of altered plastics. Alterations can occur due to a range of factors, such as exposure to heat, light, or chemicals, and can change the chemical composition and properties of the plastic. This means that the reference spectrum for an altered plastic may not match any spectrum in the database, making identification challenging [112]. In cases where the plastic is not altered, the FTIR method can provide statistics about the fitting of the spectrum obtained to

the reference. This means that the operator can define a level of acceptance based on how closely the spectrum matches the reference. For example, if there is some doubt about the identification, the operator may set a level of acceptance where the statistic is better than 75% of the matching score. Overall, while the FTIR method is a powerful tool for plastic identification, it has limitations when it comes to identifying altered plastics, and operators should be aware of these limitations when interpreting results [19].

### 6.3. Scanning Electron Microscope SEM/EDX Spectrometer

Scanning electron microscopy (SEM) and energy-dispersive X-ray spectroscopy (EDX) are valuable techniques in microplastic studies, providing insights into the morphology and elemental composition of microplastics in environmental samples. However, there are both advantages and disadvantages to using these techniques. One advantage of SEM is its high-resolution imaging capability, which can reveal the surface features and morphology of microplastics. SEM can also produce three-dimensional images of microplastics, allowing for a more accurate assessment of their shape and size. EDX, when used in combination with SEM, can provide additional information on the elemental composition of microplastics, enabling researchers to identify and quantify different types of microplastics. Another advantage of SEM and EDX is that they are relatively easy to use and require minimal sample preparation, making them accessible to many researchers. Additionally, they can be used to analyze a wide range of sample types, including sediments [88,90,91,114,115], water [88,116], salt [83], and biota [88,117], allowing for a comprehensive assessment of microplastic pollution. However, there are also limitations to using SEM and EDX in microplastic studies.

One limitation is that SEM requires a relatively large amount of sample material, which can be challenging when analyzing microplastics that are present in small quantities. Additionally, SEM and EDX cannot distinguish between different types of microplastics based solely on their elemental composition, as many types of microplastics have similar elemental profiles. Furthermore, SEM and EDX can be prone to producing false-positive results, particularly when analyzing complex samples with a high level of background noise or interference from other environmental factors. The techniques may also require specialized training and expertise to ensure accurate interpretation of the results. Overall, while SEM and EDX are valuable techniques in microplastics studies, they should be used in conjunction with other analytical methods to obtain a more complete understanding of microplastics in environmental samples. It is important for researchers to be aware of the limitations of these techniques when interpreting the results, and to use multiple lines of evidence to validate their findings [118].

### 6.4. Raman Spectroscopy

Microplastic particles can be identified in a number of environmental samples using Raman spectroscopy [3,114,118,119]. Raman spectroscopy is compatible with microscopy and utilized to investigate big, optically sorted microplastic particles. Micro-Raman spectroscopy can therefore find a large variety of size categories, from extremely tiny microplastics to those less than 1 µm [3]. Microplastic polymer categories and particles in sediment samples are frequently identified using Raman spectroscopy [120–125], in which the absorption spectrum is captured between 200 and 3500 cm$^{-1}$. In comparison to FTIR spectroscopy, Raman spectroscopy has a better lateral resolution (1 mm vs. 20 mm), a greater spectral range, a very distinctive signature spectrum, and much less liquid interference. The weakness of Raman scattering, which necessitates significantly longer acquisition times to attain a good signal-to-noise ratio, is a disadvantage of Raman spectroscopy. Raman microscopy (<20 µm), utilized in order to characterize MPs, is constrained by poor signals, but this can be solved by lengthening measurement times and fluorescent interference based on properties of materials such as color, biofouling, and deterioration [126]. Nano-Raman spectroscopy is a powerful tool for studying microplastics due to its ability to provide chemical information at the nanoscale. This technique allows for the identification and

characterization of different types of microplastics, which is important in understanding their distribution and impact in the environment. A major advantage of nano-Raman spectroscopy is its high spatial resolution. It can provide detailed information on the chemical composition of microplastics at the submicron level, making it possible to identify even the smallest particles. This information can be used to determine the sources of microplastics, their degradation pathways, and their interactions with the environment [127].

## 7. Global and Indian Scenarios of MP Pollutants in Marine and Terrestrial Environments

### 7.1. Global Scenario of MP Abundance and Distribution

Microplastic pollution differs regionally. Environmental and human variables are the main determinants of microplastic quantity and distribution [128–131]. Furthermore, it is possible that environmental variables, rather than manmade ones, are more important for the dispersion of microplastics [132,133]. The spread of microplastics is determined by natural conditions, such as wave current flow, tides, storms, wind directions, and stream hydrodynamics [134,135]. On the other hand, anthropogenic sources are human actions that cause environmental buildup of plastic litter. This study presents the global concentration and distribution of microplastics based on the various units used to describe the quantity of microplastic particles. The necessity for a uniform unit of microplastic quantity is another issue that is dependent on the procedures employed. This issue demands immediate attention, as the bulk of research has been published very rapidly. Thus, the concentration of microplastics cannot be matched across investigations [136,137]. This presents a significant challenge in comparing the amount of microplastic pollution between research and, consequently, between different regions, species, or even people. Microplastics in the sediments, water, and salt were monitored using a variety of sampling methods. The observed abundance is frequently expressed in various units. While it is occasionally possible to compare research using straightforward conversion, this is frequently not possible or involves making assumptions that result in biased conclusions. This lack of uniformity in measurement can lead to confusion and inconsistencies in data interpretation, which can hinder efforts to address the problem of microplastic pollution. Overestimation or underestimation of microplastic quantities can also have significant consequences. Overestimation can lead to unnecessary alarmism and panic, which can result in misallocation of resources and ineffective solutions. On the other hand, underestimation can lead to complacency and a lack of urgency in addressing the problem, which can perpetuate the issue and lead to further environmental damage. Therefore, a standardized unit of measurement for microplastics is essential for accurately assessing the scale and impact of microplastic pollution, identifying sources and pathways of microplastic contamination, and developing effective strategies to mitigate and prevent microplastic pollution.

The lake waters and sediments of Simcoe in Ontario, Canada contain the first discovery of microplastic particles caused by manmade activities. Sediments and surface waters were collected from eight locations using low-volume grabs and manta trawls to evaluate the type and quantity of microparticles. Moreover, the surface water grab sample had concentration levels of microplastic particles ranging from 0 to 1 particles/L, the manta trawl sample had quantities of 0–1 items/m$^3$, and soil samples had amounts of microplastic particles ranging from 8 to 1070 items/kg [135]. Isobe et al. (2016) stated that MPs such as microbeads were discovered in Hong Kong and Japanese marine water [138] and Felismino et al. (2021) found an astounding quantity of 342.2 billion microbeads in Hong Kong [135]. Veerasingam et al. (2016), have found pellets are major kind of microplastic particles, have been found on Goa beaches in India. The southern coast of Goa had far more (1150) pellet particles than the northern coast, with approximately 505 pellet particles [78]. Microplastic particles were more prevalent during the rainfall season than during dry weather. In the South Korean Nakdong River, the concentration of microplastics jumped between 260 to 1410 items/m$^3$ during the summer months from 210 to 15,560 items/m$^3$ during the wet season [139]. Nur Hazimah and Obbard studied

microplastic particles in Singapore (2014). Obbard et al. (2006) examined the microplastic amounts between 2014 and 2006, which were greater than the microplastic quantity observed in 2006, which were found to be 12–63 items/kg of dry sediment in 2014 and 0–16 items/kg of dry sediments in 2006 [140]. In research by Fauziah et al. (2015) on the prevalence of microplastic particles on Peninsular Malaysian coastlines, 265 items/m$^2$ of microplastics, including polystyrene (PS) and PE, were found [141]. Wang et al. (2017) conducted a study in Beijing River and found lot of microplastic pollutants microplastics—178 to 544 items/kg [142]. Microplastic particle concentrations in the Rhine and Main rivers of Germany were recorded as 228–3763 items/kg and 786–1368 items/kg, respectively [77]. In Northern Tibet, Lake Hovsgol, as well as the lakes in Siling, which are known for their low density of residents and minimal human activity, were analyzed, and the abundance of microplastics ranged between 14 and 1219 items/m$^2$ [54]. Su et al. investigated microplastic particles in China's most developed parts. Taihu Lake was found to have high levels of microplastic particles, with surface water concentrations reaching 3–26 items/L and dry sediment quantities reaching 11–235 items/kg. Cellophane and fibers with a size range of 100–1000 μm made up the majority of microplastics, while PE, terephthalate acid, polyester (PES), and PP were also present [143]. Peng et al. (2018) evaluated the existence of microplastic particles in river sediment samples from Shanghai, the largest urban region in China. They discovered a mean concentration of microplastics of up to $802 \pm 594$ items/kg of dry sediment, with polypropylene being the most prevalent polymer [144]. Bei Lake in central China has the highest concentration of microplastics, with quantities varying from 1660 to 8925 items/m$^3$ [144], as investigated by Wang et al. (2017). Free et al. (2014) stated that the Laurentian Great Lakes and Lake Huron, both of which are more populated, have a substantially lower microplastic concentration of 20,264 items/km$^2$ [145]. In Australia, Ziajahromi et al. (2017) found that treated wastewater revealed averages of 0.28, 0.48, and 2 items/L of microplastic effluent. PET fibers and PE with irregular shapes, which are typically seen in synthetic apparel and personal care items, were the most prevalent polymer types found [146]. Owing to contamination from a large human population and numerous enterprises, Italy's quantity of microplastics demonstrated increased concentrations in landward areas. Microplastic quantities are inversely correlated with human density, activities, urban development, tourism, and anthropogenic factors [36,146–148]. Coastal sediments in South Africa ranged from 341 to 4757 items/m$^2$ [149]. A lower concentration of microplastic particles, ranging from 204.5 to 1491.7 items/m$^3$, was found in surface water by Nel and Froneman (2015). Kanhai et al. (2017) found microplastic particles in offshore Namibia: eight items/m$^3$ [150]. The quantity of microplastic particles in the Bloukrans River was investigated by Nel et al. (2018) and compared to the summer ($6.3 \pm 4.3$ items/kg): higher levels of microplastic were found in the winter ($160 \pm 139$ items/kg) [56]. Between 2005 and 2014, the average concentration of microplastic particles in the surface water of Greenland grew by one to two items/m$^3$ [151]. In 10 g of sediment samples from Huatulco Bay, there were 2–69 microplastic particles [61]. Microplastics were reportedly found at 1–805 items/m$^2$ on the coastlines of the southeast Pacific coast [152]. According to Lusher et al. (2014), there are two items/m$^3$ of microplastic particles in the northeast Atlantic Ocean [153]. On Portugal's western coast, 66 items/m$^3$ were recorded, and the Atlantic Ocean has an estimated plastic particle concentration of 1 to 2 items/m$^3$, with most of the microplastic particles made of PE and PA [150]. Obbard et al. (2014) investigated microplastics in the Arctic Ocean region's ice cores and abundance of microplastic particles ranging from 38 to 234 items/m$^3$. The materials identified as microplastics included polyamide, polypropylene, polystyrene, and polyethylene [140]. Microplastics were identified in seawater samples taken from 18 locations in the northwest Pacific Ocean, with an average concentration of 104 items/km$^2$. Polyethylene, polypropylene, and nylon were the three types of microplastics discovered [154].

Tibbetts et al. (2018) undertook a microplastic investigation of the River Tame and four of its tributary branches, which run via Birmingham, UK, a densely populated watershed. Microplastic particles were found to be present in every sediment sample, with a mean

concentration of 165 items/kg [155]. The Tanchon stream, among the river systems feeding through the Han River in Korea, was the site of Park et al.'s (2020) investigation into the dispersion of microplastic particles in surface water, fish, and sediment. They discovered that microplastic particle concentration levels in water ranged spatial and temporal, varying around 5 and 87 items/$m^3$ (31 ± 28 items/$m^3$) [156]. Kosuth et al. (2018) have examined 159 tap water samples from throughout the world, 12 brand names of Laurentian Lakes beer, and 12 manufacturers of commercialized sea salt for the occurrence of anthropogenic particles. In the tap water, it was discovered that 81% of the microplastic particles were manmade. They claimed that each year, the average individual consumes more than 5800 particles of artificial waste from these three sources, with tap water accounting for the vast majority (88%) [157]. Galloway et al. (2017) found that the majority of microplastic particles are released into the sea, as one of the principal factors of microplastic particles in the ocean is the terrestrial source [158].

According to Zheng et al. (2020), Jiaozhou Harbour on the Shandong coastline in northern China has an abundant supply of microplastic particles ranging from 2 items/kg d.w. to 28 items/kg d.w. The MPs of various polymeric types, including PMMA, PE, PA, PET, PU, PP, and PS, have been identified through FTIR spectroscopy analysis. These particles have a size range of 0.1 to 5 mm [159]. The amount of microplastic particles in Sanggou Bay, China, ranged from 1674 to 526 items/kg. Light microscopy was used to quantify and capture the images of the particles. Five different types of polymers, including PE, PP, PS, CL, and PC, were recognized using FTIR spectroscopy [160]. The concentration of microplastic particles in Laizhou Bay, China, varied between 462 ± 167 and 193 ± 1053 particles/kg. Microplastic particles have been discovered in mixtures of PE glycol adipate, CP, PET, PP, PVA, PPA, and PVC [161]. According to Zhu et al. (2020), the Bohai Sea coastal region contained roughly 77% of all microplastic particles. The amount of microplastic particles usually ranges from 459 to 150 items/kg. Employing FTIR spectroscopy, they distinguished between the seven polymer materials ABS, PE, PP, PA, PET, PS, and rayon [162]. As per Li et al. (2019), there were 2249 items/kg of microplastic particles, or roughly 75% of the total quantity of microplastic 3 mm in size detected in southern China. SEM and energy-dispersive X-ray spectroscopy disclosed the presence of microplastics consisting mainly of PP, PE, and PS [163]. In North Carolina, Dodsona et al. (2020) observed a microplastic content of 1410 ± 810 items/kg. The study region contained 93.91% microplastic particles, which were between 5 and 0.5 mm in size. Raman microspectroscopy was used to detect the constituents of microplastic particles, which included PE, PS, NY, PP, and TEP [120]. Leads and Weinstein et al. (2019) reported that the amount of microplastics in the Charleston Harbor Estuary in South Carolina, USA was 652 items/$m^2$. In addition, the study found that 26% of particles in the study area were microplastics. ATR-FTIR was used to identify the composition of the microplastic fibers [164]. Gray et al. (2018) reported that the amount of microplastics found in the marine sediments of South Carolina estuaries such as Charleston Harbor, Winyah Bay were within the ranges of 77–414 and 26–221 items/$m^2$, respectively. The four polymer varieties of nylon, PEST, PE, and PP were distinguished using SEM [165]. As seen by Ballent et al. (2016), the nearshore of Lake Ontario has a microplastic quantity of 500 items/kg. In the research region, the size distribution of microplastic particles was 2 mm. Raman and XRF spectroscopy were used to determine the microplastic internal structure of PMMA, PDMS, PU, ABS, PS, and NY [137]. Figure 4 shows the abundance and spatial distribution of microplastics in terrestrial sediments of Europe, Asia, Africa, and South and North America. Table 3 shows the worldwide distribution and abundance of microplastics in terrestrial and marine habitats in developing and developed countries.

Table 3. Worldwide distribution and abundance of MPs in terrestrial and marine habitats in developed and developing countries.

| Developed/Developing | Country | Sample Location | MP Concentration | Sample Type | Polymer Types | References |
|---|---|---|---|---|---|---|
| Developed | China | Bohai Sea | 103 to 163 items/kg | Beach sediments | PE, LDPE, HDPE, PP, PET, PS. | [166] |
| Developed | China | Changjiang Estuary | 20–340 items/kg | Sediments | Rayon, AC, PET, PES, PS | [167] |
| Developed | Hong Kong | Coastal beaches | 49–279 items/kg | Beach sediments | LDPE, HDPE, PP | [168] |
| Developed | China | Shanghai | 802 ± 594 items/kg | Sediment | PP, PE, rayon, cotton | [144] |
| Developed | China | Three Gorges Reservoir | 25–300 items/kg | Sediment | PE, PP, PS | [169] |
| Developed | China | Beijiang River | 178 to 544 items/kg | Sediment | PE, PP, copolymer, paint particle | [142] |
| Developed | China | Taihu Lake | 11–235 items/kg | Sediment | CP, PET, PE, PA, PP | [143] |
| Developed | Italy | Sicily | 160 ± 31 items/kg | Sediment | NA | [123] |
| Developed | Italy | Lido di Dante | 1512 ± 187 items/kg | Sediment | NA | [123] |
| Developed | Italy | Lagoon of Venice | 2175 to 672 items/kg | Sediment | PE and PP | [123] |
| Developed | Italy | San Mauro | 84 ± 12 items/kg | Sediment | NA | [123] |
| Developed | Spain | Denia | 156 ± 29 items/kg | Sediment | NA | [123] |
| Developed | Spain | Barcelona | 148 ± 23 items/kg | Sediment | NA | [123] |
| Developed | France | Cassis | 124 ± 36 items/kg | Sediment | NA | [123] |
| Developed | Greece | Pilion | 232 ± 93 items/kg | Sediment | NA | [123] |
| Developed | Israel | Tel Aviv | 168 ± 16 items/kg | Sediment | NA | [123] |
| Developed | Balkan Peninsula | Bosnia | 76 ± 13 items/kg | Sediment | NA | [123] |
| Developed | Iceland | Vik | 792 ± 128 items/kg | Sediment | NA | [123] |
| Developed | Norway | Smøla | 92 ± 21 items/kg | Sediment | NA | [123] |
| Developed | Portugal | Porto | 140 ± 26 items/kg | Sediment | NA | [123] |
| Developed | Portugal | Algarve | 18 items/kg | Sediment | Rayon and PP | [170] |
| Developed | Germany | Baltic Coast | 0–7 items/kg | Sediment | NA | [57] |
| Developed | Norway | Tromsø | 72 ± 24 items/kg | Sediment | NA | [123] |
| Developed | France | Normandy | 156 ± 29 items/kg | Sediment | NA | [123] |
| Developed | Netherlands | Rottumeroog | 124 ± 27 items/kg | Sediment | NA | [123] |
| Developed | Norway | Drøbak | 100 ± 21 items/kg | Sediment | NA | [123] |
| Developed | Lithuania | Klaipéda | 700 ± 296 items/kg | Sediment | NA | [123] |
| Developed | Denmark | Fyns Hoved | 164 ± 21 items/kg | Sediment | NA | [123] |
| Developed | Slovenia | Slovenian Coast | 178 items/kg | Sediment | NA | [171] |
| Developed | UK | Scapa Flow, Orkney | 730 and 2300 items/kg | Sediment | NA | [172] |
| Developed | UK | River Thames Basin | 660 items/kg | Sediment | PP, PES, PET, PS, PE, PVC | [58] |
| Developed | UK | Birmingham | 250–300 items/kg | Sediment | NA | [173] |
| Developed | Netherlands | Dutch | 650 items/kg | Sewage sludge | NA | [174] |
| Developed | Netherlands | Meuse River | 1400 items/kg | Sediment | NA | [174] |
| Developed | Italy | Lake Bolsena | 112 items/kg | Sediment | PE, PP, PET, PVC | [175] |
| Developed | Italy | Lake Chiusi | 234 items/kg | Sediment | PE, PP, PET, PVC | [175] |
| Developed | Sweden | Lysekil | $8360 \pm 0.98 \times 10^{-3}$ items/kg | Sewage sludge | PE, PP | [60] |
| Developed | USA | Cape Hatteras Seashore | 123–196 items/kg | Beach sediments | PET, RY | [176] |
| Developed | USA | Fort Sumter Monument | 306–443 items/kg | Beach sediments | PET, RY | [176] |
| Developed | USA | Timucuan Ecological Reserve | 196–253 items/kg | Beach sediments | PET, RY | [176] |
| Developed | USA | Padre Island Seashore | 306–443 items/kg | Beach sediments | PET, RY | [176] |

**Table 3.** *Cont.*

| Developed/Developing | Country | Sample Location | MP Concentration | Sample Type | Polymer Types | References |
|---|---|---|---|---|---|---|
| Developed | USA | Buck Island Reef Monument | 56–123 items/kg | Beach sediments | PET, RY | [176] |
| Developed | USA | Dry Tortugas Park | 43–56 items/kg | Beach sediments | PET, RY | [176] |
| Developed | USA | Gulf Islands Seashore | 253–306 items/kg | Beach sediments | PET, RY | [176] |
| Developed | Canada | Ontario Lake | 980 items/kg | Lacustrine sediment | PE, PS, PU, PP, PVC, PET, PMMA | [177] |
| Developed | South Korea | Soya Island | $46,334 \pm 71,291$ items/m$^2$ | Surface water | EPS, PP, PE | [129] |
| Developed | South Korea | Heungnam Beach | $976 \pm 405$ items/m$^2$ | Beach sediment | PS | [178] |
| Developed | South Korea | Nakdong River Estuary | 27,606 items/m$^2$ | Beach Sediments | PS | [179] |
| Developed | China | Guangdong Province | 6701 items/m$^2$ | Beach Sediments | PS, PP, PP + PP/EPR | [108] |
| Developed | Hong Kong | Pearl River | 5595 items/m$^2$ | Beach Sediments | EPS | [180] |
| Developed | Japan | Japanese Sea | 1.72 million items/km$^2$ | Surface water | NA | [138] |
| Developed | China | Hanjiang River | $2933 \pm 305$ items/m$^3$ | Water | PA, PE, PET, PP, PS | [181] |
| Developed | China | Yangtze River | $2517 \pm 912$ items/m$^3$ | Water | PA, PE, PET, PP, PS | [181] |
| Developed | China | Nan Lake | $5745 \pm 901$ items/m$^3$ | Water | PA, PE, PET, PP, PS | [181] |
| Developed | Australia | Yarra River | 158 items/month | Water | PS, CP | [182] |
| Developed | Australia | Maribyrnong River | 122 items/month | Water | PS, CP | [182] |
| Developed | Russia | Kaliningrad | $1.3 \pm 0.8$ to $36 \pm 57$ items/kg | Beach sediment | Pellets, films, fibers, fragments. | [183] |
| Developing | Turkey | Dikili | $248 \pm 47$ items/kg | Sediment | NA | [123] |
| Developing | India | Ganga river | 107 to 410 items/kg | Sediment | PET, PE, PP, PS | [184] |
| Developing | India | Chennai | $439 \pm 172$ to $119 \pm 72$ items/kg | Sediment | PE, PP, Nylon (NY), PS, and PES | [19] |
| Developing | India | Rameswaram | 403 items | Sediment | PP, PE, PS, NY, PVC | [107] |
| Developing | India | Dhanushkodi | 45 to 181 items/kg | Sediment | PE, PET, PS, PP, PVC | [109] |
| Developing | India | Tuticorin | 47 to 179 items/kg | Sediment | PE, PET, PS, PP, PVC | [88] |
| Developing | India | Tuticorin, Gulf of Mannar | 8 to 17 items/kg | Sediment | PE, PP, PES, PA and paint. | [88] |
| Developing | India | Andaman | $414 \pm 87$ items/kg | Sediment | PP, PVC | [108] |
| Developing | India | Port Blair Bay, Andaman Island | $45 \pm 25$ items/kg | Sediment | NY, PU, PVC | [103] |
| Developing | India | Andaman and Nicobar | 73 to 151 items/kg | Sediment | PE, PP | [30] |
| Developing | India | Kanyakumari | 2 to 11 items/L | Water | PET, PA | [116] |
| Developing | Bangladesh | Maheshkhali | $78 \pm 9$ to $137 \pm 22$ items/kg | Sea salt | PET, PP, PE, PS | [87] |
| Developing | India | Maharashtra | 49 to 39 items/kg | Salt | PE, PET, PS, PES, PA | [86] |
| Developing | Maldives | Coral Island | $647 \pm 720$ items/m$^2$ | Beach Sediments | PP, PE, PS | [128] |
| Developing | India | Mumbai | 10–180 items/m$^2$ | Beach Sediments | NA | [94] |
| Developing | Ecuador | Guayas Province | 940 items | Intertidal sediment | LDPE, HDPE, PET, PVC, PP, PS | [185] |

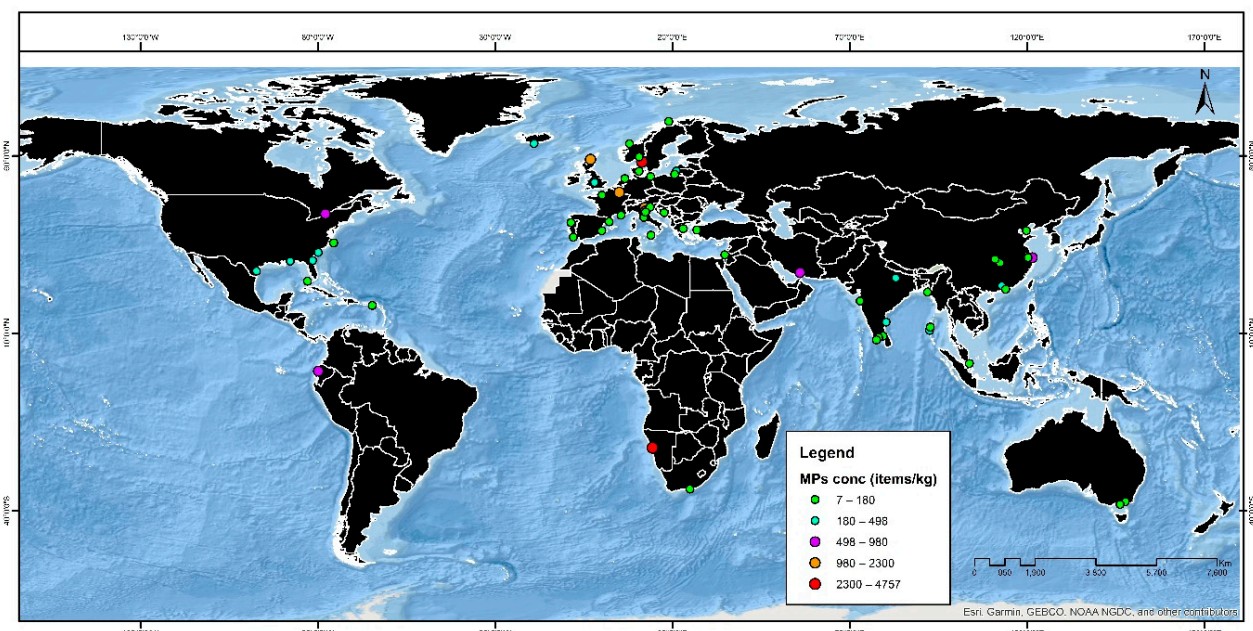

**Figure 4.** Abundance and distribution of microplastics in the terrestrial sediments of Asia, Europe, Africa, and South and North America. MP concentrations are indicated by colored circles.

### 7.2. Indian Scenario of MP Abundance and Distribution

In India, there are wide variations in the microplastic particle content of water, sediment, and salt samples. It should be observed that the concentration of MPs in the sediment samples of the far-flung Andaman Nicobar Island is around $973 \pm 77$ items/kg [110] of the Bay of Bengal was greater than that discovered in the river system of Ganga (409.86 items/kg) and the city of Chennai ($439 \pm 172$ items/kg), both located in the Bay of Bengal [186]. In Mumbai [123], microplastic concentration of $220 \pm 50$ items/kg was observed. Studies on the concentration of microplastics on India's east and west coasts have revealed that the results obtained for microplastics are expressed in various units (e.g., items/L, items/km$^2$, items/m$^3$, and percentages) depending on the sampling techniques used. Consequently, correlating these statistics is challenging. The existence of microplastic particles in the surface water along the coast of Chennai (2–11 items/L) [116] is lower than that of Tuticorin ($12 \pm 3$ to $31 \pm 2$ items/L) [88]. On the west coast of India [86], compared to the east coast [83], the quantity of microplastic particles in sea salts is significantly greater ($56 \pm 49$ to $103 \pm 39$ items/kg). The research gathered sediments from harbors, fishing activities, residential coastlines, visitor beaches, and untouched coastal areas in Kanyakumari and retrieved 343 microplastic particles from eight stations by assessing 50 g dry sediment (d.s.) in each sample location. They were mostly secondary microplastic particles, and harbors (99 particles/50 g d.s.) and popular beaches had the highest concentrations [186]. Low-density PE-type microplastic particles in sediment were found to dominate Vembanad Lake, with a concentration of 252 items/m$^2$ [187]. Maharana et al. (2019) found the abundance of microplastic particles to be $346 \pm 2$ items/m$^2$ on the west coast of India. Polymeric materials such as PE and PP, which are constituents of microplastics, were identified using FTIR technology [95].

According to Tiwari et al. (2019), the quantities of microplastics in the sandy beaches from Mumbai, Tuticorin, and Dhanushkodi were four items/kg, three items/kg, and one item/kg, respectively. These microplastic particles ranged in size from 36 μm to 5 mm. PE, PET, PS, PP, and PVC are the microplastic product categories noted, and the research region showed a comparable pattern to those categories [109]. Jayasiri et al. (2013 found that sediments of Mumbai beaches had more than 75% plastics in the 1–20 mm size range, with a concentration of 7 and 69 items/m$^2$. The notion that coastal regions are used for a variety of recreational, spiritual, fishery, and economic pursuits is a significant aspect of

the pollution problem, demonstrating the significant contribution of terrestrial sources to plastic contamination in these coastal locations [94]. Krishnakumar et al. (2020) found that microplastics are distributed in 70.9% of the Andaman and Nicobar Archipelago islands and have a content of PE, PE, and sizes ranging from 0.45 μm to 0.1 mm. The category of microplastics includes particles as small as 5 mm [30]. Karthik et al. (2018) found that sediment along the southeastern coast of India had a microplastic particle concentration of $1323 \pm 1228$ items/m$^2$, with sizes ranging between 4.75 to 9.5 mm, and a distribution of 70.9% in the research area. FTIR was used to figure out the constituents of plastic particles, which included PE, PP, and PS [188]. Imhof et al. (2017) discovered that 96% of all microplastics were between 1 and 5 mm in size. On the distant reef islands of the Maldives in the Indian Ocean, microplastic amounts range from $1029 \pm 1134$ items/m$^2$ [128]. This review makes it clear that microplastics are widespread over the world. As a result, these microplastic pollutants are unavoidable.

## 8. Microplastic Toxicity Pathways

Groundwater and the surface of terrestrial and marine habitats both include various forms of microplastics. Most of the time, ingesting contaminated food, water, or cosmetics introduces microplastic pollutants into our biological system. Less than 1 mm in size, the microplastics have been found to have the ability to absorb into the human biological system. Ingestion is the main route by which microplastics enter living organisms [158]. Once microplastic-contaminated water or food is consumed, it can pass through the mucociliary barrier and enter the digestive tract, where it may cause oxidative stress, cytotoxicity, homeostasis, immune system dysfunction, neurodegenerative disorders, inflammation, alteration of metabolism, and modification of the composition of the gut microbes [19,158].

### 8.1. Oxidative Stress, Cytotoxicity, and Homeostasis

Excessive antioxidant reactions can lead to oxidative stress, which is its main cause. Owing to their large surface area, the discharge of oxidized species that have been adsorbed to them (such as metals), or reactive oxygen compounds that are produced after an inflammatory response, microplastics may be the cause of this oxidative stress [189,190]. For example, oxidative stress following microplastic consumption has been observed in rodents and zebrafish [191,192]. Cytotoxicity is caused by such factors as particulate toxicity, oxidative stress, and inflammation. In studies involving macrophage and erythrocyte cultures, it has been observed that polystyrene induces internalization of microplastics by cells [192–194]. MPs are not membrane-bound within the cell and may interact with intercellular structures [193]. Pieces of plastic gathered from the environment have been shown to be harmful in vitro experiments [194]. Human brain and epithelial cell exposure to polystyrene and polyethylene did not result in cytolysis, but did raise reactive oxygen species (ROS) to high quantities, which contributed to cytotoxicity [195]. Consuming microplastics might have the reverse impact, leading to increased food consumed in response to elevated energy needs or decreased absorption effectiveness, as was seen in rats [192]. Moreover, the increased energy consumption caused by microplastics can result in a negative energy balance. Metabolic alterations may be brought on by microplastics, either directly or indirectly, through other impacts such as a negative energy balance. For instance, when fish and rodents are exposed to microplastic particles, lactate dehydrogenase (LDH), an anaerobic enzyme, is increased [196]. Additionally, it results in lower ATP levels and fat metabolism in the liver of rodents [157,196]. The intake of microplastics may have identical effects in people by altering metabolism, raising calorie expenditure, or lowering nutritional intake. However, because humans require more energy than the examined species and are exposed to lower quantities of radiation, it may be difficult to observe these consequences in humans. The ratio of energy available from consumption and reserves to energy expenditure affects energy homeostasis. According to several studies, microplastic particles can affect energy homeostasis. Microplastics, for example, can dramatically reduce consumption due to lower feeding activity [197–199] and lower predatory performance, possibly

due to neurotoxicity and deficiencies in digestive abilities through the manipulation of intestinal enzyme activities, with a substantial reduction in nutrient intake [196].

*8.2. Transport of Microplastics to Tissue*

Microplastic particles have the ability to relocate and affect remote tissues. If there is inflammation present, the likelihood of transfer increases due to the increased permeability of epithelial barriers. Furthermore, consumption of diets high in saturated fats and sugars, as well as malnutrition, can modify the gut microbiome and result in greater permeability of the gastrointestinal mucosa [200]. Oral dose, inhalation, and translocation of microplastic particles have been shown in rodents, reaching faraway tissues such the liver as well as the circulation [201,202]. According to a human placenta perfusion model, it has been shown that particles made of polystyrene with a size as tiny as 240 nm are capable of crossing the placenta [203]. The presence of microplastics in the bloodstream can lead to various harmful effects, including inflammation, pulmonary hypertension, vascular blockages, heightened blood-clotting ability, and the destruction of blood cells [204–206]. The potential for microplastics to accumulate in tissues and transfer up the food chain raises concerns about the impact on human health. One study estimated that people can ingest up to 5 g of plastic per week through food and water, which may have negative health effects [13,206]. The presence of microplastics in tissues can have a range of adverse health effects, including inflammation, tissue damage, and disruption of cellular processes. The release of chemicals and toxins from the plastic particles can further exacerbate these effects. The potential for microplastics to accumulate in tissues and transfer up the food chain highlights the need for continued research to better understand the potential health impacts of these particles on both wildlife and human populations.

*8.3. Immune System Dysfunction, Neurodegenerative Disorders, and Neurotoxicity*

Microplastics have the potential to elicit either local or systemic immune reactions based on their extent of spread postexposure. Nonetheless, in certain situations, such as genetic susceptibility, mere exposure to the surroundings can impair immune function and induce autoimmune disorders. The inhalation of particulate matter can activate immune cells, produce autoantibodies, and expose individuals to self-antigens by means of particle translocation, oxidative stress, and release of immune modulators, thereby leading to autoimmune diseases [207].

Exposure to microplastic pollutants can lead to neurological illnesses with neurotoxic effects. Evidence from in vivo studies indicates that particulate matter exposure may cause neuronal damage through oxidative stress, activation of the brain's immune cells (microglia), and interaction with translocated particles. [208]. According to Ranft et al. (2009), contact with traffic pollution, particularly pollutants, has been linked to dementia occurrence and moderate cognitive problems in aged people [209]. It is also linked to an increased risk of developing Alzheimer's [210]. It has also been noted that exposure to polystyrene has negative effects on neuronal activity in rats, including altered neurotransmitter levels and elevated activity of the enzyme acetylcholinesterase [192]. Understanding how microplastics might be involved in neurodegeneration in humans and increase the risk of development of neurodegenerative disorders is necessary in light of the evidence of neurotoxicity found when testing microplastics in organisms and resulting from exposure of humans to MP particulates. Figure 5 shows the toxicity pathways of microplastics in the human body. Figure 6 shows the sources of the microplastic pollutants in our environment.

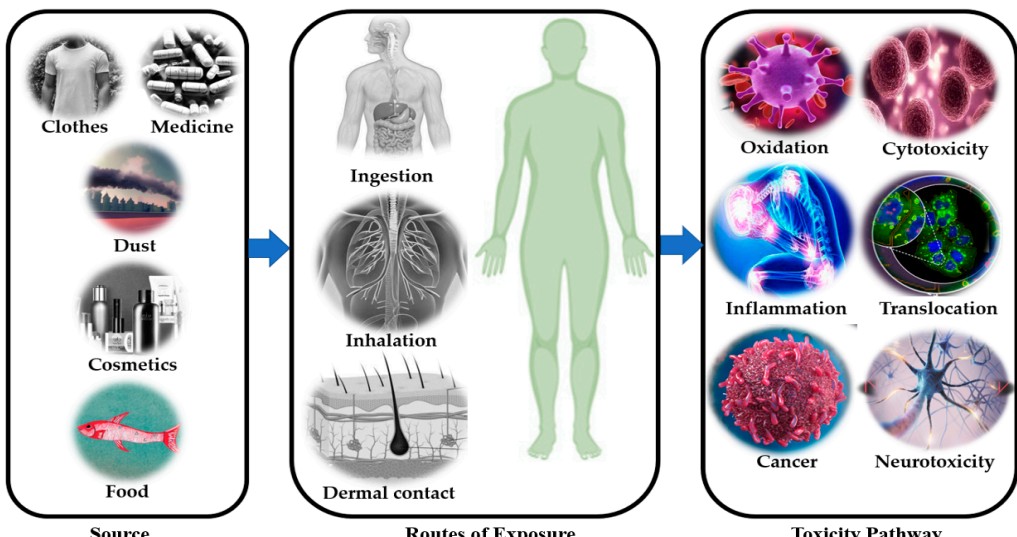

**Figure 5.** Toxicity pathways for microplastics in the human body.

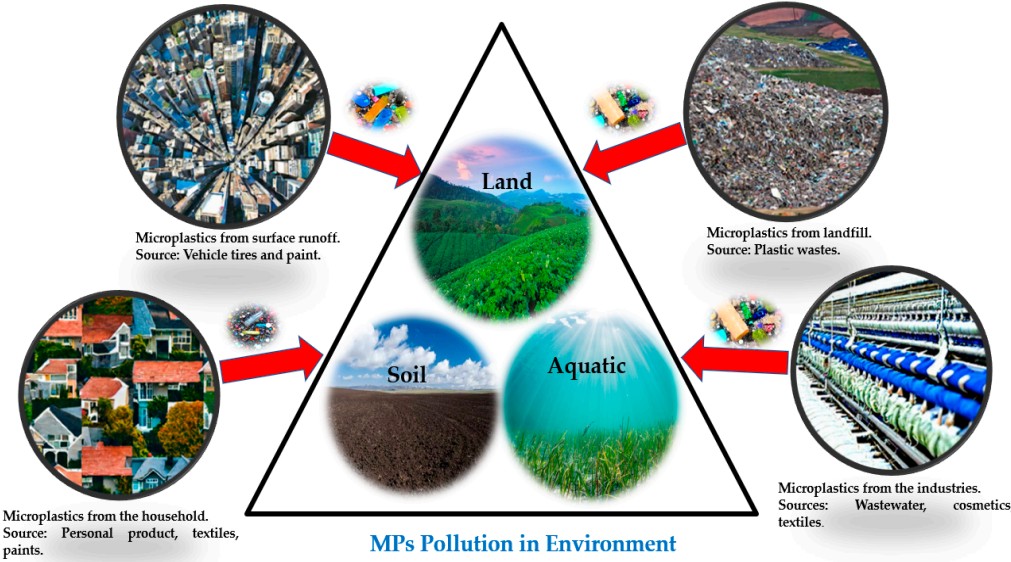

**Figure 6.** Microplastic-source pollution routes.

## 9. Remedial Methods

It is essential to concentrate on source control, which is typically accomplished through legislation, awareness campaigns, remediation, and cleanup, to remove microplastics that are already in the environment. To reduce microplastic release, there are some legal restrictions on their use. US law prohibited the use of spent microplastics beads in cosmetic goods in 2017. Many other territories, such as Australia, Canada, and the European Union (EU), are considering putting effective safeguards into place in the same way. The most significant source of primary microplastics (MPs) in the environment is the MPs used in cosmetic products. There are significant efforts being made to limit the purchase and usage of single-use plastic products, with an emphasis on plastic straws and plastic bags [211]. The consumption of single-use plastic packaging has been restricted in several countries [212]. As an illustration, the EU has suggested a Europe-wide plastics policy in news releases as support for the shift to more recycling and reuse. The usage of single-use plastics will decrease, microplastics will only be used for purposes in certain circumstances, and all plastic containers sold in the EU market will be recyclable by 2030 [213]. The EU has implemented a plastics policy aimed at reducing the impact of plastic waste on the environment. This policy includes several measures, such as reducing the consumption of single-use plastics,

promoting the use of sustainable alternatives, and improving waste management practices. While this policy is innovative and has the potential to be effective, it is not without its challenges [213]. One concern is the affordability of sustainable alternatives to plastic products. While these alternatives may be more environmentally friendly, they can also be more expensive to produce and purchase. This could make them less accessible to consumers and businesses, particularly small and medium-sized enterprises. Another challenge is ensuring that the policy is efficient in reducing plastic waste without causing unintended consequences. For example, banning certain types of single-use plastics could lead to an increase in the use of alternative materials that may have their own negative environmental impacts. It is also important to ensure that the policy does not have a negative impact on marine life and organisms [214]. To be efficient, these legislative reforms require public participation. The program's combined tax and ban on the use of plastic shopping bags is an effort to inform the people and raise their knowledge of the negative environmental effects of using plastic bags. Along with a significant increase in the utilization of remedial technologies to lessen plastic waste in aquatic habitats, there is a limitation on the production and use of microplastics [215–217]. The latter is built around floating systems that swoop down and seize plastics. Within five years, it is anticipated that this will be able to eliminate half of the plastics in the Great Pacific Garbage Patch [218]. Engineering tools, the usage of biodegradable plastics, and biotechnological instruments are the three main categories of remediation technology. The first includes cutting-edge drinking water and wastewater treatment systems. In the latter, plastics currently existing in the ecosystem are biodegraded using bacteria. To secure a decrease in microplastic contamination, an EU news release further emphasized the necessity of fostering innovation and investment in these areas [213].

### 9.1. Physical Technology in Elimination of MP Contaminant

The ecosystem receives microliters of MPs from various sources, such as textile industries, urban runoff, washing of synthetic fabrics, fishing activities, and landfill sites. These facilities offer an opportunity to create and apply cutting-edge technology to control microplastic contamination. Numerous studies have compared the ability of traditional and cutting-edge WWTP systems to eliminate plastics [215–217]. Several investigations have demonstrated that a significant portion (between 90% and 98%) of microplastics present in industrial effluents, municipal wastewater, textile effluent, and landfill leachate is removed by typical WWTP treatments [215,216]. Because of the substantial amount of wastewater emitted continuously, effluents are a cause of microplastics in water habitats [216]. Membrane deposition, electrodeposition, and coagulation are the three most popular advanced wastewater treatment methods. Among the most promising is the membrane bioreactor (MBR), which combines a membrane procedure, such as microfiltration, with a biological wastewater treatment system. Comparing microplastic removal effectiveness of this method to the overall standard activated sludge-based procedure demonstrated a higher extraction efficiency of 99% [217]. A comparison was made between the effectiveness of a membrane bioreactor method that employs microfiltration to remove microplastics and WWTPs that utilize activated sludge as the final stage [218]. The membrane technology, which discharges 1 MPs/L, has a higher separation ratio of 99%. MBRs are also contrasted with other cutting-edge tertiary treatments that offer removal efficiencies of MPs >95% from both primary and secondary microplastic pollutants, such as the rapid gravity sand filter technique. Additionally, employing artificially induced wastewater containing PE microbeads at various concentrations, the effectiveness of electrocoagulation (EC), a well-established and proven procedure for microplastic elimination from wastewater, has been investigated. Observations of microbead removal efficiencies greater than 90% under various circumstances indicate that electrocoagulation is a useful technique for eliminating microplastics from wastewater. A maximum removal efficiency of 99% was observed at pH 7.5 [187]. Washing-machine effluent is among the main sources of microplastics and fibers in municipal and surface water bodies. By utilizing activated (Ti/Pt) or inactive

BDD (boron-doped diamond) anodes and Ti cathodes in an electrochemical reactor system, it is possible to reduce the impact of these microplastic pollutants before they are discharged into the sewage system. Compared with Ti/Pt, the BDD anode displayed a clearly discernible trend and a greater removal rate. The combination of sand filters and active carbon filters can effectively remove microplastics from industrial water plants. The sand filter removes larger microplastics, while the active carbon filter removes smaller microplastics [219]. This dual-filtration approach ensures a higher level of microplastic removal and reduces the risk of these particles passing through the treatment process and into the environment [219]. According to one study, the use of sand filters in combination with active carbon filters can achieve a high level of microplastic removal. The study found that the dual-filtration approach was effective in removing microplastics with a size range of 0.1–5 mm, with an overall removal efficiency of up to 99% [220].

Recognizing how microplastics, with their tiny size and low density, will behave throughout existing drinking water treatment techniques is clearly needed, as microplastics have increasingly been found in freshwater ecosystems. However, only their behavior in PE removal using routinely used coagulants and ultrafiltration membranes has been extensively studied up to this point [121]. Techniques such as coagulation and ultrafiltration can be used to remove microplastics from drinking water supplies. The research was only conducted in a laboratory setting, but both methods worked well in a larger setting. Taking all this knowledge into account, it became clear how sophisticated final-phase water treatment systems can assist in removing microplastic contaminants from freshwater and drinking water, while also significantly reducing the microplastic pollutants discharged by WWTPs into aquatic habitats.

### 9.2. Utilization of Degradable Biopolymers

Organic raw ingredients such as starch, cellulose, lignin, and bioethanol are used to create bioplastic materials. Biodegradable plastics currently account for 0.5% of the 335 million tons of plastic generated annually [221,222]. Nonbiodegradable materials that are biobased or partly biobased include biobased PE, PP, or PET and biobased technical performance polymers such as polytrimethylene terephthalate. Although generated from biological entities, these biopolymers are just as durable as those made from petroleum. Bioplastics that can biologically degrade through aerobic or anaerobic processes are known as biodegradable polymers [223,224]. These plastics can biodegrade; however, they require favorable environmental factors and microorganisms that are not always dependent on environmental settings, which is a huge challenge. This highlights the need to define the context in which microbial degradation occurs. Certain compostable bioplastics can be decomposed by microbes into nutrient-rich biomaterials for as little as three months and do not leave any pollutants behind. There are several advantageous uses for compost, including enhancing and enriching the soil. The European Standard that establishes specific guidelines to differentiate between materials that can be considered compostable and those that cannot, as well as what qualifies as biodegradable [225]. The term "compostable" is now more commonly used than "biodegradable" due to its definability through evidence [225].

### 9.3. Bioengineered Approaches

The bioengineering-based approach involves searching for novel biodegradation pathways for conventional plastics, such as various bacterial and fungal species, or isolating the relevant enzymes to guarantee the enzymatic hydrolysis of plastic products. Intracellular carboxylesterases can solubilize biobased polyesters [226]. Specialized bacteria can decompose plastics. For example, the Ideonella sakaiensis bacterium produces two enzymes, PETase and MHETase, which work together to break down PET into its constituent parts. PETase is responsible for breaking down the PET into shorter chains of molecules called monomers, while MHETase breaks down a molecule called MHET, which is an intermediate product of PET degradation. The monomers and MHET are then metabolized by the bacterium as a source of carbon and energy. This process of PET degradation by Ideonella

sakaiensis is still not fully understood, but it has the potential to offer a solution to the problem of plastic waste in the environment [227]. Zalerion maritimum is a marine fungus that has been found to have the ability to degrade polyethylene (PE) microplastics. This fungus produces enzymes called esterases, which can break down the chemical bonds in PE, causing the plastic to disintegrate [228]. Remediation of macro- and microplastics is still in the initial stages of development at the laboratory level, although the solutions appear promising. Developing methods for the in situ microbial degradation of MPs through microbial addition or improved natural attenuation employing local microflora is essential. Therefore, in order to be acceptable for widespread application, bioengineering-based technologies must undergo extensive additional studies, and development is required.

*9.4. Phytoremediation, Biochar, and Air Filters*

Microplastic contamination is a growing concern in the environment, including in soils and the atmosphere. Various remediation technologies have been developed to address this issue. One approach to remediate microplastic contamination in soil is through the use of plant-based phytoremediation. This involves the use of plants to take up and degrade microplastics in the soil. For example, the plant species Arabidopsis thaliana has been shown to absorb and degrade polyethylene microplastics in the soil through the activity of enzymes produced by the plant [102,229]. Other plant species, such as clover and ryegrass, have also been found to have the ability to remove microplastics from soil [230]. Recent studies have demonstrated the potential of phytoremediation for the removal of microplastics from contaminated soils and waters. For example, a study found that the water hyacinth plant was effective in removing microplastics from aqueous solutions, with removal rates of up to 93% [231]. Another study found that the duckweed plant was effective in removing microplastics from soil, with removal rates of up to 83% [232]. Another remediation technology for microplastic contamination in soil is the use of biochar. Biochar is a type of charcoal produced from biomass that can be added to soil to improve soil quality and remove pollutants. Studies have shown that biochar can effectively remove microplastics from soil by adsorption and can also enhance soil microbial activity and improve soil structure [233]. One study found that adding biochar to contaminated soil reduced the concentration of microplastics by up to 60% [234]. Another study found that biochar reduced the bioavailability of microplastics in soil, which could reduce their potential to harm plants and other organisms [235].

In the atmosphere, the most common remediation technology for microplastic contamination is air filtration. Air filters can be used to capture microplastic particles and prevent them from entering indoor and outdoor environments. However, it is important to note that the efficiency of air filters in capturing microplastics can vary depending on the type and size of microplastics present in the air [235]. The efficiency of air filters in removing microplastics from the atmosphere has been a topic of research in recent years. One study found that high-efficiency particulate air (HEPA) filters were effective in removing microplastics from indoor air, with a removal efficiency of up to 95% [236]. Another study found that electrostatic air filters were also effective in removing microplastics from the air, with a removal efficiency of up to 90% [237]. Plant-based phytoremediation and the use of biochar are promising remediation technologies for microplastic contamination in soil. Air filtration is a common remediation technology for microplastic contamination in the atmosphere, but its efficiency may vary depending on the type and size of microplastics present in the air.

## 10. Awareness, Management, and Mitigation Strategies

The physiological parameters of humans and other organisms are affected by the presence of microplastics in their surroundings, resulting in poor well-being. Plastic consumption must be constantly tracked on global and national scales. To help understand the different processes that govern the abundance of plastic particles in our ecosystems and their physiological effects on organisms, scientific studies are still required. A public

understanding of the negative effects of microplastics in this area is urgently needed [238]. This would inspire a range of suggestions for reducing the consumption and utilization of plastic materials and their by-products. The recommendations of decision-makers for eliminating plastic pollution, including limiting, reusing, and recycling plastic wastes, manufacturing and usage limitations, use of biobased and compostable plastic waste [15], waste disposal system improvements, and eco-friendly design, must be incorporated into management and mitigation strategies. The EU has taken a leading role in the regulation of microplastics. In 2019, the EU adopted a restriction on intentionally added microplastics in products such as cosmetics, detergents, and cleaning products, which are a significant source of microplastic pollution. The regulation requires that these products contain no more than 0.01% microplastics by weight. The EU also aims to establish a harmonized methodology for monitoring microplastic pollution in the marine environment to assess the effectiveness of mitigation strategies [239]. Similarly, in the United States, the Microbead-Free Waters Act was passed in 2015, banning the production and sale of personal care products containing microbeads, which are a type of microplastic [240]. Canada has also taken action on microplastic pollution by listing microbeads as a toxic substance under the Canadian Environmental Protection Act. In addition, the Canadian government has implemented a ban on the sale of toiletries containing microbeads [241]. Mitigation strategies to address microplastic pollution include source reduction, which involves reducing the production and use of microplastic-containing products. Recycling and waste management strategies can also be implemented to prevent microplastics from entering the environment. For example, the adoption of circular economy principles can help reduce plastic waste by keeping materials in use and out of the environment.

## 11. Conclusions

Microplastic contamination has grown into a global problem, but researchers still lack adequate monitoring, toxicity, analysis, and remediation technology. As an outcome, the distribution, toxicity, analytical method, health risk, and remediation technologies of microplastics were examined in this review paper, with the findings being summed up as follows.

- Industrial wastewater, litter, wastewater treatment plants, domestic and personal products, city runoff, and fishing waste are some of the pollutant pathways for microplastics, which eventually find their way into terrestrial and aquatic environments to contaminate aquatic life and be ingested by people.
- Microplastic analysis was divided into sampling, pretreatment, and analytical sections. The sample section covered water, sediment, and salt samples. In the pretreatment phase, the handling of density-difference analytical techniques and techniques for potential applications in microplastic analysis are thorough separation, and the best way to remove other impurities in addition to microplastics was thoroughly addressed. The majority of currently employed microplastic removal strategies are outlined in the extraction section.
- Engineering approaches, biopolymers, and biological technologies have been used to summarize various methods for microplastic cleanup.
- Governments are addressing the microplastic issue, and in the upcoming years, we may expect to see more initiatives aimed at preventing pollution, such as restrictions on the use of plastic bottles, bags, and many other plastic products. The European Union made a press announcement stating that by 2030, all plastic packaging sold in the EU market must be capable of being recycled, as per the recent development.
- To compile reliable information for future strategies and planning for the mitigation of microplastic contamination, uniform guidelines are required.

In conclusion, there is widespread microplastic pollution that negatively impacts human life. Microplastics (MPs) have been found in remote regions, such as the Arctic and mountain lakes, due to atmospheric pathways. Wind can transport MPs over long distances, and settle on remote areas such as snow and ice caps or mountain peaks. Sources

of microplastics in the atmosphere are diverse, including road dust, agricultural practices, and industrial emissions. Recommendations to reduce microplastic pollution include using eco-friendly alternatives, such as natural fiber clothing and biodegradable packaging, implementing strict regulations on plastic production and waste management, and educating the public on the harmful effects of microplastics. As a result, there is an immediate need for increased ongoing research among scientists, as well as national strategies to combat microplastic pollution. Research on microplastic pollution, assessment, surveillance, and removal technologies is currently insufficient.

**Author Contributions:** Conceptualization, V.N.P. and T.S.; methodology, V.N.P. and T.S.; software, V.N.P. and E.K.; resources, T.S. and P.L.; data curation, V.N.P. and E.K.; writing—original draft preparation, V.N.P. and T.S.; writing—review and editing, E.K. and P.L.; visualization, T.S. and U.S.; supervision, T.S. and P.L. All authors have read and agreed to the published version of the manuscript.

**Funding:** This research received no external funding.

**Data Availability Statement:** Not applicable.

**Conflicts of Interest:** The authors declare no conflict of interest.

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
