# Peer review of "A Global Perspective on Microplastic Occurrence in Sediments and Water with a Special Focus on Sources, Analytical Techniques, Health Risks, and Remediation Technologies"

_water, doi:10.3390/w15111987_

Round 1

Reviewer 1 Report

Water 2354149

Dear editor,

The paper is informative and well written. The authors have produced a Review Paper, which has well summarized the information on the given topic at a very reasonable level. The article is well systematized and has a good coverage of cited literature. We believe that the work is useful to the scientific public and that it is necessary to publish it after subsequent changes.

Illustrations are needed. They should be completely changed:

1. The size of letters and numbers should be the size of the surrounding text

2. The choice of pictures and illustrations should be completely clear and obvious

3. The resolution is poor

The units throughout the text you use are different. Reduce them to basic units (SI). For example, use only m, m2, m3 and not cm, cm2, cm3, ..L. Use only g not kg...

Be aware of abbreviations. You have not introduced MP anywhere. Use either only abbreviations or the name itself.

It is not clear what it is  END, (STP)?

Use one abbreviation for:  micro - plastics, microplastics, MPs, MP particles

Somewhere you use FT-IR and somewhere you use FTIR

Somewhere you use (SEM)/EDX and somewhere SEM/EDX

Somewhere you use MPs/kg, microplastics/m2, particles/m2 do you also use this unit?

items/m2, pieces/m2

MPs L−1

mg per person/day

Table 1 is confusing. Correct table.

I suggest that you split Table 3 into two or three tables with comparable data. Dream the labels through the table that load, like items/kg, or sediment

Article contains many different writing styles. Sentence constructions and the use of different terms for the same thing. Improve the articles to the same better level of writing.

Reviewer 2 Report

Dear authors

Your manuscript is of great interest for the scientific community. It can be very usefull for people concerned by microplastic situation and you bring a nice and quite complete overview about these materials.

I suggest you to consider some of the reviewers remarks to make your manuscript better and more attractive. For these reasons I consider that it can be published after minor revisions

Best regards

Reviewer 3 Report

Dear Authors, 

As per my view the manuscript is falling under the scope of the journal and of great interest to the national, regional and global researcher working in the field of microplastic toxicity and its impact on humankinds.  Overall, paper written well and need some short of corrections/improvement, kindly see my suggestions below: 

Line no 37-39: need to be supported with the references (Env. Sci. Pollut. Res.  29(54); https://doi.org.10.1007/s11356-022-23929-2; Water 2023, 15, 51. https://doi.org/10.3390/w15010051) 

In the data collection section kindly discussed with the number of articles were collected using those keywords and how you segregated the articles for use 

Table 3: if authors can categories the countries into (Southeast Asia, Europe, USA.....) or (developed / developing countries) will be of great interest for regional researchers 

Resolution of Figure 5 need to be increased to make the presentation clear 

Section " Management, and Mitigation strategies" need to be strengthen by discussing some policies and regulations (kindly see  Microplastics in terrestrial ecosystems: Un-ignorable impacts on soil characterises, nutrient storage and its cycling. TrAC Trends in Analytical Chemistry, 116869. https://doi.org/10.1016/j.trac.2022.116869) 

Line no 791-792: confusing ? what quick action ?

Conclusion is written well but if authors could give some short of recommendations in 2-3 lines will be great to the young scientist, as this point is very much important
